# The Role of Exosomes in Cancer Progression

**DOI:** 10.3390/ijms23010008

**Published:** 2021-12-21

**Authors:** Beáta Soltész, Gergely Buglyó, Nikolett Németh, Melinda Szilágyi, Ondrej Pös, Tomas Szemes, István Balogh, Bálint Nagy

**Affiliations:** 1Department of Human Genetics, Faculty of Medicine, University of Debrecen, Egyetem tér 1, H-4032 Debrecen, Hungary; gbuglyo@hotmail.com (G.B.); nemeth.nikolett@med.unideb.hu (N.N.); szilagyi.melinda@med.unideb.hu (M.S.); balogh@med.unideb.hu (I.B.); nagy.balint@med.unideb.hu (B.N.); 2Geneton Ltd., 841 04 Bratislava, Slovakia; ondrejpos.sk@gmail.com (O.P.); tomasszemes@gmail.com (T.S.); 3Comenius University Science Park, Comenius University, 841 04 Bratislava, Slovakia; 4Division of Clinical Genetics, Department of Laboratory Medicine, Faculty of Medicine, University of Debrecen, H-4032 Debrecen, Hungary

**Keywords:** liquid biopsy, exosomes, biomarkers, cell-free nucleic acids, cancer

## Abstract

Early detection, characterization and monitoring of cancer are possible by using extracellular vesicles (EVs) isolated from non-invasively obtained liquid biopsy samples. They play a role in intercellular communication contributing to cell growth, differentiation and survival, thereby affecting the formation of tumor microenvironments and causing metastases. EVs were discovered more than seventy years ago. They have been tested recently as tools of drug delivery to treat cancer. Here we give a brief review on extracellular vesicles, exosomes, microvesicles and apoptotic bodies. Exosomes play an important role by carrying extracellular nucleic acids (DNA, RNA) in cell-to-cell communication causing tumor and metastasis development. We discuss the role of extracellular vesicles in the pathogenesis of cancer and their practical application in the early diagnosis, follow up, and next-generation treatment of cancer patients.

## 1. Introduction

Extracellular vesicles (EVs) were discovered in 1946 as procoagulant platelet-derived particles in ultracentrifuged pellets from blood plasma [1]; however, the term “extracellular vesicle” was first used later in independent observations [2,3]. We now know that EVs are nano- to micron-sized vesicles covered by phospholipid bilayers. They are classified into three groups: apoptotic bodies (ABs), ectosomes/microparticles (MPs)/microvesicles (MVs), and exosomes (Exs) [4]. Their subtypes include large dense core vesicles, membrane blebs, oncosomes, outer membrane vesicles (OMVs), and prostasomes (exosome-like vesicles) [5,6,7]. According to current nomenclature, MVB-derived EVs are called exosomes, while plasma membrane-derived EVs are microvesicles [8,9]. The contents, membrane composition and size of EVs are highly heterogenous and depend on cellular location and environment. Extracellular vesicles are released by different tissues and cell types. They are found in body fluids, including amniotic fluid, ascites, bile, breast milk, nasal and bronchial lavage fluid, blood plasma, saliva, semen, synovial fluid, and urine, allowing extraction of EVs from various liquid biopsies [6,7]. EVs contain proteins, lipids, DNA, RNA, and microRNA serving as mediators of cell-to-cell communication [10,11,12]. Membranes protect their contents from nuclease and protease degradation and micro-environment changes (e.g., osmolarity and fluctuations in pH) [12]. Apoptotic bodies are released by blebbing of plasma membrane during apoptosis. The second main group of EVs includes vesicles of different size and components. The last main group, exosomes, belongs to the class of intraluminal vesicles (ILVs) contained in multi-vesicular bodies (MVBs) and are released to the extracellular environment by MVBs fusing with the cell membrane [7,13].

EVs are potent vehicles of intercellular communication providing protection and maintenance for cells and regulating cellular functions [6]. They are involved in cell-to-cell communication, immune response, angiogenesis, and signal transduction [14]. Many cancer cells release EVs, thereby affecting tumor microenvironments and suppressing or, surprisingly, stimulating immune responses, leading to a delicate balance of immune modulation [15]. Tumor-derived EVs were shown to avoid perforin-mediated elimination by CD8+ lymphocytes, a function in which their adenosine content seems to play a pivotal role [16].

Several techniques are available for the isolation of EVs, including density-gradient centrifugation (sucrose and iodixanol gradients), filtration, precipitation, size-exclusion chromatography, and ultracentrifugation; however, subgroups of EVs are not easy to distinguish due to an absence of specific markers. Identification of contents may help (e.g., by antibody-coupled bead flow cytometry analysis, electron microscopy supplemented with immunogold staining method and immunoblotting). Some tetraspanins (CD9, CD63 and CD81) were identified as specific markers for exosomes, but were later noted in the other groups [17,18]. Some other molecules were also determined as markers for EVs (14-3-3 proteins, MHC molecules, stress proteins (HSP), tumor susceptibility gene 101 (TSG101) and ESCRT-3 binding protein ALIX [19]), but CD63 and TSG101 were observed in all EV groups in a comprehensive cancer study showing different distributions based on their appearance and origin [20]. The solution leading to a reliable classification system may be to analyze the glycol-pattern of EVs (shown to be altered from that of the parent cell membrane [21]) as glycosylation of glycans is different between exosomes and apoptotic bodies [22].

In this review, we discuss the current understanding of exosomes’ role in cancer, and as markers of disease progression, making them valuable assets in tumor diagnosis and treatment.

## 2. Liquid Biopsy

Liquid biopsy has many advantages over conventional, invasive methods: it is less invasive, easily obtainable and repeatable. Liquid biopsies may serve as sources of many important biomarkers including cancer cells, extracellular vesicles (apoptotic bodies, microvesicles and exosomes), tumor-educated platelets, metabolites, proteins, and cell-free nucleic acids (cf-DNA, cf-RNA). Cf-DNAs are used in prenatal testing and characterization of the mutation profile of tumor cells. The number of publications on liquid biopsy-derived exosomes used for cancer detection and monitoring have skyrocketed recently, even with the problems of classification and standardizing extraction methods for different biofluids [23].

## 3. Extracellular Vesicles

### 3.1. Apoptotic Bodies

Apoptotic bodies (ABs) were discovered in 1972 by Kerr et al. [24]. They are formed from cells undergoing chromatin condensation, followed by membrane blebbing and fragmentation of cellular components during apoptosis. Finally, ABs are cleared when their translocated phosphatidyl serine membrane components bind to the Annexin V receptor of phagocytes [25], and the C3b complement or thrombospondin is recognized by and bound to phagocytes [26]. The apoptotic death of a cell breaks it up into a variable number of ABs, which are the largest vesicles among EVs (ranging from 1000 nm to 5000 nm) [27]. Apoptotic bodies contain large amounts of RNA both short and long [28], but other macromolecules (DNA, lipids, and proteins) were also observed in them [18,29]. Phosphatidylserine is a useful marker of ABs [30] (Table 1).

### 3.2. Microvesicles

Microvesicles (MVs) are another class of EVs, first described in 1967 as “dust” from platelets [31]. Their importance became obvious in the past decades. Microvesicles range from 50–200 nm up to 1 µm in diameter [18,32,33] and contain a portion of the plasma membrane separated by blebbing outward and released into the extracellular space [12]. MVs are found in the blood [34] and in other body fluids, e.g., urine [35], cerebrospinal fluid [36], tears [37], saliva [38] and nasal secretions [39], ascites [40], and semen [41]. They are involved in intercellular communication, signaling pathways and promotion of cell invasion by cell-independent matrix proteolysis. Ectosomes, microparticles, oncosomes, shedding bodies, and shedding vesicles are all referred to as microvesicles [42,43].

Tumor-derived microvesicles (TMVs) and oncosomes originate from cancer cells [44] and an altered release of MVs is associated with cancer progression. Higher numbers of MVs indicate a more severe disease, and a high amount of proteolytic content in MVs correlates with the quick spreading of breast cancer and fibrosarcoma [45]. TMVs transfer bioactive molecules (nucleic acids, lipids, and proteins) to recipient cells causing disease, promoting cancer and providing diagnostic markers. In TMVs, the membrane-type 1 matrix metalloprotease (MT1-MMP) promotes cell invasion of the extracellular matrix (ECM) [46].

Some specific markers might be identified as MV markers, e.g., adenosine diphosphate ribosylation factor 6, CD40 ligand, various integrins and selectins [32,47,48,49]. ARF6 is involved in tumor formation, growth, and metastasis [47] (Table 1).

### 3.3. Exosomes

Exosomes were first reported as a “type of small vesicles” in Pan and Johnstone’s sheep reticulocyte-related experiment in 1983 [50], and the name “exosome” was first used (and visually described using transmission microscopy) by Johnstone et al. in 1989 [51]. Among the groups of EVs, exosomes are the smallest (40–100 nm) [52]. They may be verified based on their cup-shaped morphology by negative-staining transmission microscopy and the presence of markers (CD9, CD63, and CD81) [53,54,55]. Besides tetraspanins, other molecules are also found on the surface of exosomes, e.g., ALIX, TSG101, HSP70, flotillin, LAMP-1, MHC-I, -II [8,12,55,56,57]. Exosomes originate as intraluminal vesicles by inward budding of endosomal membranes forming multivesicular bodies (MVBs) from internal multivesicular compartments of the endocytic pathway. Following the secretory pathway, exosomes merge with the plasma membrane and are released into the extracellular milieu or fused with lysosomes for degradation [8,9] (Figure 1).

Some studies in the 2000s proved that exosomes may deliver RNA such as mRNA and miRNA involved in cell-to-cell communication [11,54]. During their biogenesis, exosomes are packed with various cargo such as nucleic acids (e.g., DNA, mRNA, miRNA, lncRNA, circRNA) [11,58,59], proteins [60,61], lipids [62], metabolites [63,64] (Figure 1). When each component is released, they affect intercellular communication through direct cell-to-cell interaction contributing to tumorigenesis [11,58,59,60,61,62,63] and tumor-derived exosomes filled with RNAs and proteins may transfer oncogenic activity to recipient non-tumor cells [58,59,60,64] (Figure 1, Table 1).

Moreover, such tumor-derived exosomes may also serve as biomarkers of prognosis and response to therapy [65,66,67].
ijms-23-00008-t001_Table 1Table 1Characterization of extracellular vesicles from different aspects.Extracellular Vesicle TypeExosomeMicrovesicleApoptotic BodySize40–100 nm50–1000 nm1000–5000 nmPlasma/Serum Concentration5.3 particle/mL × 10^6^5–50 g/mLMuch lower compared to MVs and EXsOriginInward budding of endosomal membranes forming MVBs and then released by exocytosisOutward budding/blebbing of plasma membraneProgrammed cell death or apoptosisMode of extracellular releaseConstitutive and regulatedRegulatedRegulatedContentProteins, lipids, DNA (gDNA, mtDNA, ncDNA), mRNA, miRNA, lncRNA, circRNAProteins, lipids, mRNA, miRNA, ncRNAsNuclear fractions, cell organelles, proteins, mRNA, ncRNA, DNAMarkersALIX, TSG101, tetraspanins (CD81, CD63, CD9, CD51), HSP70, flotillin, LAMP-1, MHC-I, -IIPhosphatidylserin, Integrins, selectins, CD40, flotillin-2, metalloproteinases, tissue and cell-specific factorsAnnexin V, histones, phosphatidylserinFunctionIntracellular communicationIntracellular communicationFacilitation of phagocytosisMorphologyCup-shapeCup-shapeHeterogeneousIsolation methodsUltracentrifugation, size exclusion chromatography, chemical precipitation, peptide affinity methodCentrifugationNo standard method (Centrifugation)DetectionFlow cytometry with capture beads, electron microscopy, Western blotFlow cytometry, electron microscopy, capture-based assaysFlow cytometry, electron microscopyReference[12,55,56,57,68][12,55,57,69][12,57,70]


## 4. Exosomes in Tumors

Some studies described the connection between exosomes and the development and progression of different cancer types. Exosomes may serve as multicomponent biomarkers in tumor diagnostics [71,72].

### 4.1. Exosomes in Tumor Progression

Under physiological conditions, exosomes contribute to normal cellular function, while their altered function may cause cancer. Exosomes induce tumor development by changing the landscape of tumor microenvironments, and immune system activation by changing vascularity and cell polarity. They are also responsible for epithelial–mesenchymal transition (EMT) and an interconversion to mesenchymal–epithelial transition (MET) in several human malignancies [73]. Exosomes are involved in promoting tumorigenesis and metastasis and are associated with gaining chemoresistance, but exosomes derived from dendritic cells may be engineered to trigger antitumor immune responses (“dexosomes”) [74,75].

Exosomes have different ways to participate in tumor development. They may alter gene expression. Post-translational modifications such as ubiquitination frequently occur on EVs. Some studies showed that a dysregulation of ubiquitination and deubiquitination may lead to various diseases such as cancer, and they are part of the regulation of metabolic reprogramming in cancer cells [76]. Exosomes rich in Wnt5b have been associated with head and neck squamous cell carcinomas, invasive breast cancer, and lung and pancreatic cancer [77].

In addition to their effect on gene expression, they may be involved in posttranslational modifications as explained below. EVs from ovarian carcinoma were found to be enriched in mannose and sialic acid residues [22], i.e., glycosylation is modified.

Phosphorylation is another example: the role of Src-phosphorylation in the angiogenesis of myeloid leukemia is promoted by exosomes. This type of phosphorylation may be therapeutically targeted [78].

### 4.2. Exosomes in Cancer Immunology

Exosomes containing nucleic acids may control the innate and adaptive immune responses [74]. Secreted exosomes may stimulate the anticancer activity of effector CD4+ T cells [4] and promote their proliferation by indirectly activating naïve T cells and B cells by interacting with antigen-presenting cells [79]. Exosomes derived from various cells may release molecules which evoke immune responses in tumor formation. The role of Th17 cells is well-known in ovarian cancer: they may secrete pro- and anticancer factors, and promote angiogenesis [80,81]. Ye et al. (2014) reported that exosomes from nasopharyngeal carcinoma (NPC) cells hindered the proliferation of T cells and the differentiation of Th1 and Th17 cells by reducing the level of interleukin-2 (IL-2), interferon gamma (IFN-γ), and interleukin-17 (IL-17). However, the same exosomes activated Treg cells as a consequence of increasing the levels of interleukin-1b (IL-1b), interleukin-6 (IL-6), and interleukin-10 (IL-10) released from T cells [82]. The Treg/Th17 ratio was elevated in primary and metastatic tumors of patients with epithelial ovarian cancer compared to benign tumors and peritoneum. Inequality of the Treg/Th17 ratio is caused by an exosome-mediated transfer of miR-29a-3p and miR-21-5p from macrophages to T helper (CD4+) cells; the process is suppressed by signal transducer and activator of transcription 3 (STAT3) signaling [83]. Macrophage-derived exosomes of liver cancer cells induced the secretion of IL-6, MCP-1, IL-10, and TNF-α via STAT3 signaling [84]. Transfer of miRNA let-7d via exosomes—as analyzed in TME cells—reduced Th1 proliferation and secretion of IFN-γ [85].

### 4.3. Exosomes in Immunosuppression

Exosomes carry programmed cell death receptor ligand 1 (PD-L1), which may directly prevent the anticancer function of CD8+ T cells in vivo [65,86], so evasion of immune surveillance may be possible by exosomes. Some authors are optimistic about the future of anti-PD-L1 therapy of cancer [87,88]. PD-L1 on the surface of extracellular vesicles is associated with immunosuppression, disease progression of tumor patients and altered response to immunotherapy [65,86].

### 4.4. Exosomes in Angiogenesis and Lymphangiogenesis

Cellular microenvironments have a major impact on cancer progression. Exosomes released by cancer cells precondition tissue environments for local spreading and distant metastasis by delivering inflammatory and other factors [89,90,91]. Exosomes from breast cancer promote adhesion of cells to extracellular matrix proteins. Exosome-induced metastases were reported by several authors. Tumor-derived exosomes may affect factors related to EMT and escaping from immune surveillance (β-catenin, caveolin-1, transforming growth factor beta (TGFβ), etc.) [91].

The growth and spread of tumors are associated with the formation of new blood and lymph vessels. Cells located in a tumor and its microenvironment secrete angiogenic factors leading to tumor angiogenesis and growth factors contributing to lymphangiogenesis. Tumor cell-derived exosomes loaded with non-coding RNAs are involved in the same processes [92,93]. Several recent studies proved that cancer cell-derived exosomal miRNAs promote angiogenesis and/or lymphangiogenesis using different signaling pathways affecting endothelial cells [94,95,96,97] and non-endothelial cells [98]. Tumor-derived exosomes containing miR-21 or let-7a (under hypoxic stress) increase M2 polarization of macrophages which may stimulate tumor-associated angiogenesis and lymphangiogenesis [99,100]. Exosomal long non-coding RNAs are also reported to be involved in the process [101,102,103]. For example, lncRNA H19 and HOTAIR stimulate angiogenesis by synthesis and secretion of vascular endothelial growth factors [104,105] and other lncRNAs are involved by sponging microRNAs [106,107]. Exosomal circRNA-100338 may regulate angiogenesis to promote metastasis in hepatocellular carcinoma [108].

Other factors delivered by exosomes may also play key roles in promoting angiogenesis. Some studies have shown that exosomes from breast cancer transfer Annexin II (a tumorigenic factor) both in vivo and in vitro [109]. New vessel formation was detected predominantly in hypoxic regions of tumors with a low level of chemoresponsiveness [110].

### 4.5. Exosomes in the Therapy Phase

Radiotherapy promotes the secretion of exosomes, and their content plays a significant role in cancer survival. Irradiation may unexpectedly increase survival of cancer cells by triggering the release of exosomes carrying survivin (an apoptosis inhibitor) [110]. It may also affect the ratio of migratory factors (such as Insulin Like Growth Factor Binding Protein 2 (IGFBP2) and exosome connective tissue growth factor (CTGF) in exosomal cargo [111]. Various forms of stress (e.g., heat and oxidative stress) and some chemotherapeutic drugs (such as proteasome inhibitors like Bortezomib and alkylating agents like melphalan) may induce and then increase exosome release from cancer cells [112,113,114].

## 5. Exosomes in Cancer Detection

Exosomes contain nucleic acids, proteins, lipids, and carbohydrates. They are suitable for the detection of different types of cancer non-invasively from liquid biopsies.

### 5.1. Nucleic Acids

Nucleic acids found in exosomes show the mutations present in the cells from which they are derived. Types include genomic/nuclear DNA (gDNA, nDNA), mitochondrial DNA (mtDNA), messenger RNA (mRNA), small non-coding RNAs such as microRNA (miRNA), PIWI-interacting RNAs, YRNAs, and long non-coding RNAs (lncRNA) including circular RNAs (circRNA). Many of them are promising biomarker candidates in the diagnosis and prognosis of cancer and monitoring of patients in early- and late-stage disease, therapy selection and follow-up [115,116,117,118,119,120,121,122,123,124,125,126,127]. Other RNAs were also detected in exosomes, such as transfer RNAs (tRNAs) and viral RNAs [128].

#### 5.1.1. DNA

##### Genomic/Nuclear DNA

In cancer patients, more gDNA content in exosomes is derived from cancer cells than from normal cells due to apoptosis or necrosis. Exosomes with nuclear content are secreted by tumor cells in high quantities, allowing for their application as cancer biomarkers. Recent studies revealed the connection between micronuclei and exosomes with nuclear content [129].

DNAs are shuffled into MVBs by tetraspanins and DNA-binding proteins interacting with CD63 [129]. Exosomes are released from MVBs containing molecules derived from them and then release cell-free DNA into plasma [130]. Circulating tumor DNA is partially derived from exosomes of tumor cells, which may serve as biomarkers in cancer diagnosis [131].

Circulating cell-free DNA encapsulated in exosomes in the plasma of gastrointestinal tumor patients may transform normal gastrointestinal cells into tumor cells, in a process known as genometastasis [132,133,134].

##### Mitochondrial DNA

Apart from genomic DNA, mtDNA is also present in exosomes and may show relevant copy number differences between cancer patients and healthy controls, as shown by our group in ovarian carcinoma [117]. Release of mtDNA in exosomes was also characteristic for cancer-associated fibroblasts in a breast cancer model [89].

#### 5.1.2. RNA

##### MicroRNAs

The role of miRNA is highlighted as a marker in diagnosis and monitoring of the progression of many types of cancers [135]. It has been shown that miR-21 is overexpressed in exosomes obtained from patients with esophageal squamous cell carcinoma and glioblastoma [136]. Fabbri and colleagues found that exosomal miR-21 and miR-29a may reduce the overall survival of lung cancer [137]. Anfossi et al. measured the level of miR-21 and found that it is a valuable diagnostic biomarker for breast cancer [138].

##### Long Non-Coding RNAs

Exosome-derived lncRNAs are emerging as useful cancer biomarkers, and peripheral blood is not the only liquid biopsy from which they are obtained—e.g., urine exosomes carry markers for urothelial bladder cancer, while cervicovaginal lavage yields exosomes containing lncRNA relevant for the diagnosis of cervical cancer [139]. A combination of 2 exosomal mRNAs and an exosomal lncRNA—breast cancer anti-estrogen resistance 4 (BCAR4)—was reported as a robust diagnostic marker set for the screening of colorectal cancer [121]. Plasma lncRNA long intergenic non-protein-coding RNA 152 (LINC00152) is thought to be present exclusively in exosomes. In the diagnosis of gastric cancer, LINC00152 was reported to be more sensitive than established markers [122]. Exosomal Metastasis-associated lung adenocarcinoma transcript 1 (MALAT-1) is overexpressed in non-small cell lung cancer and is thought to increase proliferation and migration capabilities of tumor cells [140].

##### Circular RNAs

Interestingly, exosomal circRNAs are globally downregulated in most cancer types, but individual tumor biomarkers are usually still present [123]. Their tertiary structure makes them stable and suitable for clinical use. Exosomal circ-0051443 has been reported to be underexpressed in hepatocellular carcinoma [141], while hsa_circ_0065149, circ-KIAA, and hsa_circ_0000419 were downregulated in plasma samples of gastric cancer patients [142,143,144].

### 5.2. Proteins

#### 5.2.1. Tetraspanins

Tetraspanins such as CD9, CD63, CD81 and CD151 are scaffolding membrane proteins highly enriched in exosomes [145,146]. Logozzi et al. (2009) demonstrated that plasma CD63+ exosomes are significantly increased in melanoma patients [147]. Yoshioka et al. found CD63 to be present in higher levels in exosomes produced by malignant cells, providing evidence that exosomal CD63 could be a protein marker for cancer [20]. CD81 plays a critical role in hepatitis C attachment and cell entry [148].

#### 5.2.2. SNARE Proteins

The main function of SNARE proteins composed of multiple proteins is promoting the fusion of vesicle membranes and plasma membranes [149]. The vesicle associated membrane protein 7 (VAMP7), a member of the SNARE family, is an important component of exosomes involved in their secretion to the extracellular environment [150]. The abnormal lncRNA LINC00511 induces formation of invadopodia by regulating the colocalization of VAMP7 and synaptosome associated protein 23 (SNAP23) and is thus involved in tumor progression as shown in hepatocellular carcinoma (HCC) cells [151].

#### 5.2.3. Rab Proteins

There are more than 60 different Rab proteins in humans. Rab proteins are small GTPase proteins regulating membrane trafficking, intracellular transport, lipid remodeling, fusion, and exosome release [152,153]. RAB11 was the first protein from the RAB family that was shown to be involved in the secretion of exosomes containing TFR and HSC70 from myelogenous leukemia cell lines [154]. Depending on the cell type, Rab5, Rab7, Rab11, Rab27, and Rab35 are involved in vesicle secretion and thus cancer progression. It was observed that ovarian cancer cells significantly increased their exosome release in hypoxia by upregulating Rab27a and downregulating Rab7, LAMP1/2, and NEU-1 [155].

#### 5.2.4. Annexins

Annexins are a group of calcium- and phospholipid-binding proteins highly expressed in exosomes in cancer. Maji et al. demonstrated their participation in breast cancer pathogenesis [109].

#### 5.2.5. Flotillins

Flotillins are membrane-associated proteins involved in scaffolding, signaling, and endocytosis. They are enriched in exosomes and may be used as exosomal biomarkers. Phuyal et al. showed that they affect the composition of exosomes [156].

#### 5.2.6. Proteins Involved in ESCRT Complex

Programmed cell death 6-interacting protein (PDCD6IP) or ALIX is a cytoplasmic protein involved in apoptosis as a binding protein in endosomal sorting complex required for transport (ESCRT) complexes [157]. Specific components are sorted into ILVs by ESCRT complexes (ESCRT-0, ESCRT-I, ESCRT-II, and ESCRT-III, which contain specific proteins, e.g., VPS4, VTA1, and ALIX) involved in the biogenesis of exosomes [158,159]. The hepatocyte growth factor-regulated tyrosine kinase substrate (HRS), which is part of ESCRT-0, also has a significant role in the exosome biogenesis [159]. HRS recognizes ubiquitinated proteins and interacts with STAM (a part of ESCRT-0) [160]. ALIX (ESCRT-III-related protein) may cooperate with syndecan-syntenin and is also involved in exosome formation [161].

#### 5.2.7. Heat Shock Proteins

Exosomes also contain heat shock proteins (HSPs) which are produced under stressful conditions. HSPs were found to play a role in antigen presentation by loading peptides onto the major histocompatibility complex (MHC). Thus, they have the capacity to stimulate antitumor immune responses. Lv et al. found that exosomes from resistant human hepatocellular carcinoma cells may improve tumor immunogenicity by the induction of HSP-specific natural killer (NK) cell responses [162].

### 5.3. Lipids

The lipid composition of exosomes varies and is affected by the cell of origin. Skotland et al. used exosomal lipids as biomarkers and reported that engineering lipid composition of exosomes may yield useful drug delivery vehicles [163].

### 5.4. Glycans

During glycosylation, glycans (a subgroup of saccharides) may be attached to lipids, proteins, and other glycans [164]. They are essential in exosome function [165]. Conserved and unique glycan contents were found in exosomes corresponding to parent cell membranes (e.g., specific sialic acid-containing glycoproteins in exosomes derived from SKOV3 cells) [166,167]. Bisecting GlcNAc-containing N-glycans were found in exosomes derived from SKOV3 and OVM ovarian carcinoma cell lines [167]. These molecules have been associated with a lower occurrence of metastasis in multiple cancer types [168], suggesting that exosomes containing glycans may be used in cancer vaccine development. The glycan properties of cancer cell-derived exosomes make them promising early diagnosis markers such as lectin-conjugated nanoparticles in pancreatic cancer [169].

Nucleic acid, protein, lipid, and carbohydrate contents of exosomes are summarized in Table 2.

## 6. Exosomes as Next-Generation Treatment Options

Recently, exosome research has been focused on developing designed drug or nucleic acid delivery to cancerous cells or tissues to improve the effectiveness of cancer therapy [170,171,172,173,174,175,176,177,178,179,180,181,182,183,184,185,186,187,188,189,190,191,192] (Table 3).

Useful properties of exosomes (small size, compatibility with biological processes, long circulatory halflife, compliance to adaptation and modification, enhanced permeability and retention (EPR) effect, prolonged circulation, tumor-targeting capacity) make them promising therapeutic shuttle vesicles [193,194,195,196,197] with low toxicity following administration [198]. They may be used to transport anticancer drugs, biomolecules, nucleic acids, soluble proteins, antibodies, and nanoparticles [199,200,201,202,203,204,205]. Exosome-based immunotherapeutics are under development and testing in animal models and clinical trials. Exosome-based clinical trials involved in cancer immunotherapy are available in the database of privately and publicly funded clinical studies (https://www.clinicaltrials.gov/, accessed on 4 December 2021).

Recent reviews highlighted the possible therapeutic application of exosomes for personalized medicine. Engineered extracellular vesicles loaded with various molecules may find application in EV-based personalized medicine as a new option for tailoring clinical treatment [206,207,208,209].

## 7. Conclusions

In this review, we summarized the role of extracellular vesicles in cancer progression focusing on exosomes carrying extracellular nucleic acids (DNA, RNA) from cell to cell, causing tumor and metastasis development. We outlined the roles of liquid-biopsy-derived exosomes in tumor pathology and therapy against cancer. They serve as excellent sources of various markers for early non-invasive detection, classification of cancer and follow up. They are suitable for the targeted delivery of drugs to tumor cells, making them promising next-generation treatment vehicles.

## Figures and Tables

**Figure 1 ijms-23-00008-f001:**
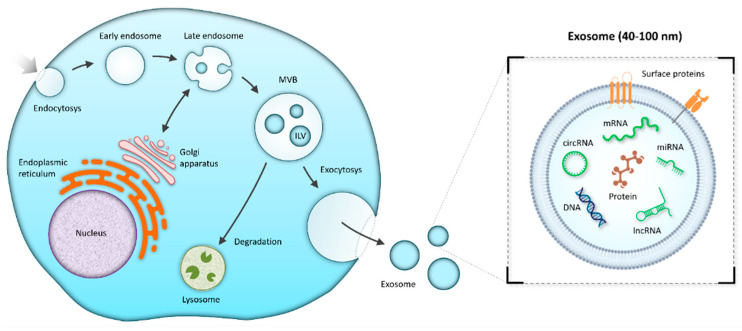
Exosome biogenesis. The process of endocytosis results in the formation of an early endosome, followed by a maturation to a late endosome that may bidirectionally exchange vesicles with the Golgi apparatus and the endoplasmic reticulum. Invagination of late endosomal membranes forms the intraluminal vesicles (ILVs) contained in the multivesicular body (MVB). The MVB may fuse with the plasma membrane and release ILVs to the extracellular space as exosomes. On the other hand, the MVB may also be transported to a lysosome for subsequent degradation of its content.

**Table 2 ijms-23-00008-t002:** A summary of exosome contents.

Type			Function	Application	Reference
Nucleic acids	DNA	gDNA/nDNA	unknown	prenatal diagnosis, biomarker	[115,116]
		mtDNA	unknown	biomarker	[117]
	RNA	mRNA	codes for proteins	data	[118]
		miRNA	gene regulation	diagnosis	[119,120]
		lncRNA	regulation of gene transcription, epigenetic modification	diagnostic biomarker	[121,122]
		circRNA	gene regulation, cell proliferation, epithelial-mesenchymal transition, metastasis, invasion, chemoresistance	diagnostic biomarker	[123]
Proteins	Tetraspanins	CD9, CD63, CD81, CD51	adhesion, proliferation, migration, binding, entrance, motility	biomarker	[145,146]
	Rab proteins	Rab5, Rab7, Rab11, Rab27 and Rab35	vesicle secretion	cancer prognosis	[155]
	SNARE proteins	e.g., VAMP7	secretion of exosomes, involved in tumor progression	monitoring the tumor progression	[150]
	Annexins		cell life cycle, exocytosis, apoptosis	cancer pathogenesis	[109]
	Flotillins		scaffolding, signaling, endocytosis	biomarker	[156]
	Heat shock proteins		antigen presentation	improving tumor immunogenicity	[162]
Lipids			formation of exosomes and releasing of exosomes to the extracellular environment	biomarker	[163]
Glycans			decrease in metastasis	possible use in cancer vaccine development	[168]

**Table 3 ijms-23-00008-t003:** Application of exosomes in cancer therapy.

Cancer	Application	Reference
Breast	Effective doxorubicin therapy using targeted iRGD-exosome delivery of doxorubicin	[170]
Breast	Exosomes loaded with miR-379 from engineered mesenchymal stem cells may reduce tumor activity	[171]
Bladder	Delivery of polo-like kinase-1 (PLK-1) siRNA containing exosomes to cancer cells decreases the PLK-1 mRNA	[172,173]
Glioma	Anti-survivin immunotherapy leads to decreased release of CD9+/GFAP+/SVN+ and CD9+/SVN+ exosomes which may be associated with longer progression-free survival	[174]
Glioma	miRNA-146b (anti-glioma miRNA) containing exosomes derived from marrow stromal cells may suppress glioma growth in vitro	[175]
Glioblastoma	Natural-killer-derived exosomes may stimulate T cell proliferation and promote the maturation of DCs	[176]
Hepatocellular carcinoma	Exosomes enriched with miR-335-5p may decrease cancer growth and invasion	[177]
Hepatocellular carcinoma	Dendritic cell-derived exosomes (DEXs) promote natural killer cell and T cell activation and proliferation	[178,179]
Leukemia	Tumor-derived exosomes (TEXs) carry tumor-associated antigens that trigger tumor antigen-specific immune response	[180]
Lymphoma	TNF-alpha-related-apoptosis-inducing-ligand (TRAIL)—armed exosomes may promote apoptosis in cancer cells	[181]
Murine Lewis lung carcinoma	Paclitaxel (PTX) loaded exosomes (exoPTX) increased cytotoxicity in cancer cells (drug resistant MDCKMDR1 (Pgp+) cells)	[182]
Murine melanoma	Macrophage-derived exosome-encapsulated Trp2 vaccine may induce a stronger antigen-specific cytotoxic T cell response via Th1 response	[183]
Nasopharyngeal carcinoma	TEXs loaded with galectin-9 suppress T-cell proliferation, and increase apoptosis in mature Th1 lymphocytes	[184,185]
Osteosarcoma	Exosomes filled with miR-101 may suppress lung metastasis in osteosarcoma	[186]
Ovarian cancer	Tumor-derived exosomes expressing Fas ligand and TRAIL induce apoptosis of the precursors of DCs and PBMCs	[187]
Pancreatic ductal adenocarcinoma	Exosomes transfected with miR-145-5p may suppress pancreatic ductal adenocarcinoma cell proliferation and invasion through TGF-β/Smad3 pathways	[188]
Prostate	Tumor exosomes expressing Fas ligand induce apoptosis of CD8 (+) T cells	[189]
Prostate	Delivery of paclitaxel from cancer cell-derived exosome increases drug cytotoxicity	[190]
Prostate	Presence of ASC-derived exosomal miR-145 initiates apoptosis in prostate cancer	[191]
Prostate	Knockdown of *ACTN4* gene decreases the invasion and proliferation of prostate cancer	[192]

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
