# Peer review of "The Role of Exosomes in Cancer Progression"

_ijms, 2021, doi:10.3390/ijms23010008_

Round 1

Reviewer 1 Report

Comment 1.

The authors should highlight the previously known mechanisms of whether the exosome-derived DNAs might be transformed into cell-free (cf) DNA in blood plasma. That is, explain the possibility that cfDNA or ctDNA may originate from exosome-derived DNAs in blood.  

Author Response

Dear Reviewer 1,

We would like to thank you for reading our manuscript, for you time and effort to improve the quality of our manuscript.

We would like to thank the Reviewer 1’s comment, and we wrote a new paragraph to this section.

“DNAs are shuffled into MVBs by tetraspanins and DNA-binding proteins interacting with CD63 [129]. Exosomes are released from MVBs containing molecules derived from them and then release cell-free DNA into plasma [130]. Circulating tumor DNA is partially derived from exosomes of tumor cells, which may serve as biomarkers in cancer diagnosis [131].

Circulating cell-free DNA encapsulated in exosomes in the plasma of gastrointestinal tumor patients may transform normal gastrointestinal cells into tumor cells, in a pro-cess known as genometastasis [132, 133, 134].”

In the revised version of our manuscript, the "Track changes" of the Microsoft Word was used to show the corrections in the text. 

Sincerely yours,

Dr. Beáta Soltész, Ph.D            

Reviewer 2 Report

The authors have done excellent job by putting together a wealth information about exosome biology and its implication in cancer progression. Some minor revisions are requested.

  1. Using a schematic for visual representation of exosome generation pathway could increase the viewership and the merit of the work.
  2. If exosome in tumor topic can be divided into small subtopics that will be easier to follow. For example demonstration of exosome in different types cancers  ( for reference, as authors described the topic : exosome in cancer detection ) 
  3. What is the role of exosome in angiogenesis and Lymphangiogenesis , as these two processes are strongly associated with regulation of tour microenvironment. 
  4. If authors can add a sub topic focusing on role of exosome in regulation of immunosuppression during cancer progression, that possibly can add value in the MS.

Author Response                                                                                                  Debrecen, 04.12.2021

Dear Reviewer 2,

Please find enclosed our corrected manuscript entitled „The role of exosomes in cancer progression”, submitted to Special Issue "Cell-Free Nucleic Acids in the Molecular Pathogenesis of Diseases Development 2.0".

In the revised version of our manuscript, the "Track changes" of the Microsoft Word was used to show the corrections in the text. 

We would like to thank the Reviewer 2’s comments, and we made a schematic to present the biogenesis of exosomes, we divided the exosome in tumor topic into subtopics, we added a new paragraphs with the title of “Exosomes in angiogenesis and lymphangiogenesis” and “Exosomes in immunosuppression”.

“Figure 1. Exosome biogenesis. The process of endocytosis results in the formation of an early endosome, following maturation to a late endosome that can bidirectionally exchange vesicles with the Golgi apparatus and the endoplasmic reticulum. Invagination of late endosomal membranes forms the intraluminal vesicles (ILVs) contained in the multivesicular body (MVB). The MVB can fuse with the plasma membrane and re-lease ILVs to the extracellular space as exosomes. On the other hand, MVB can also be transported to lysosomes for subsequent degradation of its content.”

“4.1. Exosomes in tumor progression

4.2. Exosomes in cancer immunology

4.3. Exosomes in immunosuppression

4.4. Exosomes in angiogenesis and lymphangiogenesis

4.5. Exosomes in the therapy phase”

“4.3. Exosomes in immunosuppression

Exosomes carry programmed cell death receptor ligand 1(PD-L1), which may directly prevent the anticancer function of CD8+ T cells in vivo [65, 86], so evasion of immune surveillance may be possible by exosomes. Some authors are optimistic about the future of anti-PD-L1 therapy of cancer [87, 88]. PD-L1 on the surface of extracellular vesicles is associated with immunosuppression, disease progression of tumor patients and altered response to immunotherapy [65, 86].

4.4. Exosomes in angiogenesis and lymphangiogenesis

Cellular microenvironments have a major impact on cancer progression. Exo-somes released by cancer cells precondition tissue environments for local spreading and distant metastasis by delivering inflammatory and other factors [89, 90, 91]. Exosomes from breast cancer promote adhesion of cells to extracellular matrix proteins. Exosome-induced metastases were reported by several authors. Tumor-derived exo-somes may affect factors related to EMT and escaping from immune surveillance (β-catenin, caveolin-1, transforming growth factor beta (TGFβ, etc.) [91].

The growth and spread of tumors are associated with the formation of new blood and lymph vessels. Cells located in a tumor and its microenvironment secrete angiogenic factors leading to tumor angiogenesis and growth factors contributing to lymphangiogenesis. Tumor cell-derived exosomes loaded with non-coding RNAs are involved in the same processes [92, 93]. Several recent studies proved that cancer cell-derived exosomal miRNAs promote angiogenesis and/or lymphangiogenesis using different signaling pathways affecting endothelial cells [94, 95, 96, 97] and non-endothelial cells [98]. Tumor-derived exosomes containing miR-21 or let-7a (under hypoxic stress) increase M2 polarization of macrophages which may stimulate tu-mor-associated angiogenesis and lymphangiogenesis [99, 100]. Exosomal long non-coding RNAs are also reported to be involved in the process [101, 102, 103]. For example, lncRNA H19 and HOTAIR stimulate angiogenesis by synthesis and secretion of vascular endothelial growth factors [104, 105] and other lncRNAs are involved by sponging microRNAs [106, 107]. Exosomal circRNA-100338 may regulate angiogenesis to promote metastasis in hepatocellular carcinoma [108].

Other factors delivered by exosomes may also play key roles in promoting angiogenesis. Some studies have shown that exosomes from breast cancer transfer Annexin II (a tumorigenic factor) both in vivo and in vitro [109]. New vessel formation was detected predominantly in hypoxic regions of tumors with a low level of chemoresponsiveness [110].”

We would like to thank you for reading our manuscript, for your time and effort to improve the quality of our manuscript.

Sincerely yours,

Dr. Beáta Soltész, Ph.D